

# Effect of sex on glucose handling by adipocytes isolated from rat subcutaneous, mesenteric and perigonadal adipose tissue

Floriana Rotondo[1], Ana Cecilia Ho-Palma[1], Xavier Remesar[1,2,3], José Antonio Fernández-López[1,2,3], María del Mar Romero[1,2,3] and Marià Alemany[1,2,3]

[1] Department of Biochemistry and Molecular Biomedicine, University of Barcelona, Faculty of Biology, Barcelona, Spain
[2] Institute of Biomedicine, University of Barcelona, Barcelona, Spain
[3] CIBER OBN, Centro de Investigación Biomédica en Red: Obesidad y Nutrición, Barcelona, Spain

Corresponding author
Marià Alemany, malemany@ub.edu

## ABSTRACT

**Background**. Adult rat epididymal adipocytes are able to convert large amounts of glucose to lactate and glycerol. However, fatty acid efflux is much lower than that expected from glycerol levels if they were the product of lipolysis. Use of glucose for lipogenesis is limited, in contrast with the active glycolysis-derived lactate (and other 3-carbon substrates). In this study, we analyzed whether white adipose tissue (WAT) site and sex affect these processes.

**Methods**. Mature adipocytes from perigonadal, mesenteric and subcutaneous WAT of female and male rats were isolated, and incubated with 7 or 14 mM glucose during 1 or 2 days. Glucose consumption, metabolite efflux and gene expression of glycolytic and lipogenesis-related genes were measured.

**Results**. The effects of medium initial glucose concentration were minimal on most parameters studied. Sex-induced differences that were more extensive; however, the most marked, distinct, effects between WAT sites, were dependent on the time of incubation. In general, the production of lactate was maintained during the incubation, but glycerol release rates increased with time, shifting from a largely glycolytic origin to its triacylglycerol (TAG) lipolytic release. Glycerol incorporation was concurrent with increased TAG turnover: lipolytic glycerol was selectively secreted, while most fatty acids were recycled again into TAG. Fatty acid efflux increased with incubation, but was, nevertheless, minimal compared with that of glycerol. Production of lactate and glycerol from glucose were maximal in mesenteric WAT.

**Discussion**. Female rats showed a higher adipocyte metabolic activity than males. In mesenteric WAT, gene expression (and substrate efflux) data suggested that adipocyte oxidation of pyruvate to acetyl-CoA was higher in females than in males, with enhanced return of oxaloacetate to the cytoplasm for its final conversion to lactate. WAT site differences showed marked tissue specialization-related differences. Use of glucose for lipogenesis was seriously hampered over time, when TAG turnover-related lipolysis was activated. We postulate that these mechanisms may help decrease glycaemia and fat storage, producing, instead, a higher availability of less-regulated 3-carbon substrates, used for energy elsewhere.

## INTRODUCTION

Adipose tissue is a large disperse organ, closely related to most organs, and distributed in anatomically differentiated masses, traditionally considered storage depots of fat reserves (*Cinti, 2005*). There are marked differences between these sites (*Jamdar, 1978*; *Prunet-Marcassus et al., 2006*), up to the point that their main common characteristic is the massive accumulation of fat (essentially triacylglycerols, TAG), stored in vacuoles, taking most of the cell space (*Rotondo et al., 2016*). The marked oxidative metabolism of brown adipose tissue, with multiple (and smaller) fat vacuoles and a large number of mitochondria, contrasts with the predominantly glycolytic activity of the huge single-vacuole adipocytes of white adipose tissue (WAT) (*Ho-Palma et al., 2016*). Between these extremes, we find a number of intermediate adipocyte phenotypes, such as beige (*Wu et al., 2012*) or pre-adipocytes (*Torio-Padron et al., 2010*). The array of cell types, which retain a number of similar characteristics, result in an ample variability of functions of BAT, WAT; but also beige and *brite* or brown-in-white adipose tissue (*Giralt & Villarroya, 2013*). Nevertheless, for most of the earlier research, the prevailing idea of WAT function was just that of a fat (energy) storage organ. The dangers derived from WAT (and/or individual adipocytes) hypertrophic growth are closely related to the development of metabolic syndrome (*Kaplan, 1989*), a chronic state of inflammation (*Cancello & Clément, 2006*). In the last twenty years, many other functions of WAT have been discovered, including the synthesis of hormones and growth factors, which interaction with other organs and cell types potentially affecting overall health. WAT regulatory effects are often exerted by small masses of adipose tissue, such as perivascular (*Guzik et al., 2007*), pericardial (*Greif et al., 2013*) or intramuscular (*Smith et al., 1998*).

The active presence of a large number of other cell types interspersed between adipocytes, such as stem (*Zuk et al., 2002*), immune system (*Qiu et al., 2016*), stromal vascular (*Ribeiro Silva et al., 2017*) and even hematopoietic (*Luche et al., 2015*) is seldom taken into account from the point of view of substrate handling. Nevertheless, their role is critical for regulation (*Kopecký et al., 2004*; *Rosen & Spiegelman, 2006*) and for other WAT functions. The diversity of cell proportions, functions and site distribution contribute to extend, diversify and specialize the roles of WAT in a number of roles, from the control of energy partition to defense and the control of the function of other organs (*Scherer, 2006*; *Alemany, 2011*). This is largely achieved through hormonal (*Kershaw & Flier, 2004*) and cytokine (*Wisse, 2004*) signaling.

In mammals, body fat distribution is sex-related, and is more marked in humans (*Kuk et al., 2005*) than in rodents. Essentially, the differences in WAT distribution are a consequence of differences in overall metabolic function (*Ailhaud et al., 1991*; *Fried, Lee & Karastergiou, 2015*); i.e., we have observed, recently, a marked effect of sex on rat WAT amino acid metabolism (*Arriarán et al., 2015a*; *Arriarán et al., 2015b*). We also found that

WAT, and isolated adipocytes, are able to convert large amounts of glucose into lactate and glycerol (*Arriarán et al., 2015c*; *Ho-Palma et al., 2016*), despite its small percentage of active cytoplasm with respect to tissue weight (*Rotondo et al., 2016*). This anaerobic conversion of glucose to 3-carbon substrates (3C), was not related to hypoxia, since it took place in the presence of abundant oxygen. In addition, epididymal adipocytes from adult rats converted most of $^{14}$C-labelled glucose to labelled 3C (*Ho-Palma et al., 2018*).

The analysis of how a disperse organ (*Vitali et al., 2012*) in which the different sites play different functions, could maintain its distinctive organization and uniform control was one of the critical objectives of the present study. The objective was to find out whether the site differences were a question of adipocyte metabolic specialization within a frame of metabolic capabilities and parameters that define WAT as a specific (albeit disperse) organ. Consequently, we selected three quite different WAT sites: perigonadal (PG), mesenteric (MES) and subcutaneous (SC) WAT. PG, i.e., epididymal or periovaric, contains large cells, and a high percentage of fat per g of tissue (*Romero et al., 2014*); it is considered essentially an example of storage WAT (*Ruan et al., 2003*). In rats, MES WAT is less consistent, lax, and structurally complex in its connections, in part bridging the gap between intestine (and dietary nutrients) and the liver (*Novelle et al., 2017*). In humans, MES accounts for most of "visceral" WAT. SC WAT is even more diverse, with extreme differences between locations. Subcutaneous WAT is one of the most studied adipose tissues in humans (*Gensanne et al., 2009*) because of its accessibility. We used the inguinal fat pads, which provide a clearly distinguishable (and uniform) site with sufficient material for analysis.

In brief, our main objective in carrying this study was to test whether widely different (location, structure, fat content, blood flow, relationship with other organs, and median cell size) WAT sites shared a basic web of metabolic pathways, and, alternatively, showed distinct ways to handle energy substrates. The study was focused on the known relationships of glucose with lipogenesis, glycerogenesis, and especially, glycolysis to lactate (*Sabater et al., 2014*; *Rotondo et al., 2017*). We included sex as another potentially important variable, since most of our previous work had been done using males, and the sexual differentiation of adipose tissue metabolism plays a critical role in obesity and other metabolic syndrome pathologies (*Medrikova et al., 2012*).

## MATERIALS AND METHODS

### Animals and sampling

The experimental design and the rat handling procedures were applied following the animal treatment guidelines established by the corresponding European, Spanish and Catalan Authorities. The Committee on Animal Experimentation of the University of Barcelona specifically authorized the procedures used in this study (approval number: 9443).

Wistar rats (Janvier, Le Genest-Saint Isle, France), 14-week old (eight male and eight female), were used after at least 7 days of adaptation to the new environment. The animals were kept in two-rat cages under standard conditions: i.e., 21.5–22.5 °C, and 50–60% relative humidity; lights were on from 08:00 to 20:00. The rats had free access to water and standard rat chow (#2014, Teklad Diets, Madison, WI, USA) available in excess at any time.
When the females were in the proestrus phase, the rats were weighed, and killed at the beginning of a light cycle. After complete anesthesia using isoflurane, the animals were exsanguinated with syringes by puncturing the aorta. They were rapidly dissected, excising samples of mesenteric (MES) WAT, cleaned of attachments and pancreatic tissue, epididymal /periovaric (i.e., PG) WAT, and both inguinal pads of SC WAT.

## Experimental groups

Tissue samples of a pair of same-sex rats with similar weight were coarsely minced and pooled prior to the separation of adipocytes. In all, four two-rat samples were used for adipocyte extraction of each WAT site and sex. The cells were distributed in groups, incubated independently in at least four wells each (one for each pool of two rats of the same sex): The variables to study were: (a) sex: female and male; (b) site: SC, MES and PG; (c) initial glucose concentration: 7 mM and 14 mM; and (d) time of incubation: 24 h or 48 h. In total, there were 48 groups of four different rat pools. In all cases, at the end of the incubation, the cells were harvested and used for gene expression analysis, while the media were used for analysis of glucose and metabolites as described below.

## Preparation and incubation of adipocytes

Adipocytes were isolated by incubation with collagenase (type I #LS004196, Worthington Biomedicals, Lakewood, NJ, USA), as described in a previous paper (*Rotondo et al., 2016*), essentially following the Rodbell procedure (*Rodbell, 1964*). Adipocyte numbers were counted from the final washed suspensions. Their (spherical when free) diameters were measured using serial microphotographs (Fig. 1) and the ImageJ software (http://imagej.nih.gov/ij/) (*Baviskar, 2011*). The recovery of adipocytes with respect to the mass of WAT used was estimated in a number of randomly selected samples, as previously described (*Rotondo et al., 2016*). Incubations were carried out using 12-well plates (#734-2324VWR International BVBA/Sprl., Leuven Belgium). The incubation medium consisted of 1.7 ml of DMEM (#11966-DMEM-no glucose; Gibco, Thermo-Fisher Scientific, Waltham, MA, USA), supplemented with, 30 mL/L fetal bovine serum (FBS; Gibco). The medium contained added glucose at a final nominal concentration of 7 mM or 14 mM. The medium also contained 25 mM hepes (Sigma-Aldrich, St Louis, MO, USA), 2mM glutamine (Lonza Biowhittaker, Radnor, PA, USA), 1 mM pyruvate (Gibco), 30 mg/mL delipidated bovine serum albumin (Millipore Calbiochem, Billerica, MA, USA), 100 nM adenosine, 100 mU/mL penicillin and 100 mg/L streptomycin (all from Sigma-Aldrich).

Each well contained 400 $\mu$L of the cell suspension. Since 0.1 mL of medium was used for initial measurements (metabolite zero values, cell counting), the final incubation volume was 2.0 mL. The plates were kept at 37 °C in an incubation chamber, ventilated with air supplemented with 5% $CO_2$. The cells were incubated for 24 h or 48 h, without any further intervention, as previously described (*Ho-Palma et al., 2016*; *Rotondo et al., 2016*).

The incubation of adipocytes was stopped by pipetting out the whole contents of the well, allowing the adipocytes to float and form a defined layer, which was taken out. The infranatant medium was centrifuged (to recover any remaining adipocytes), mixed, aliquoted and frozen.
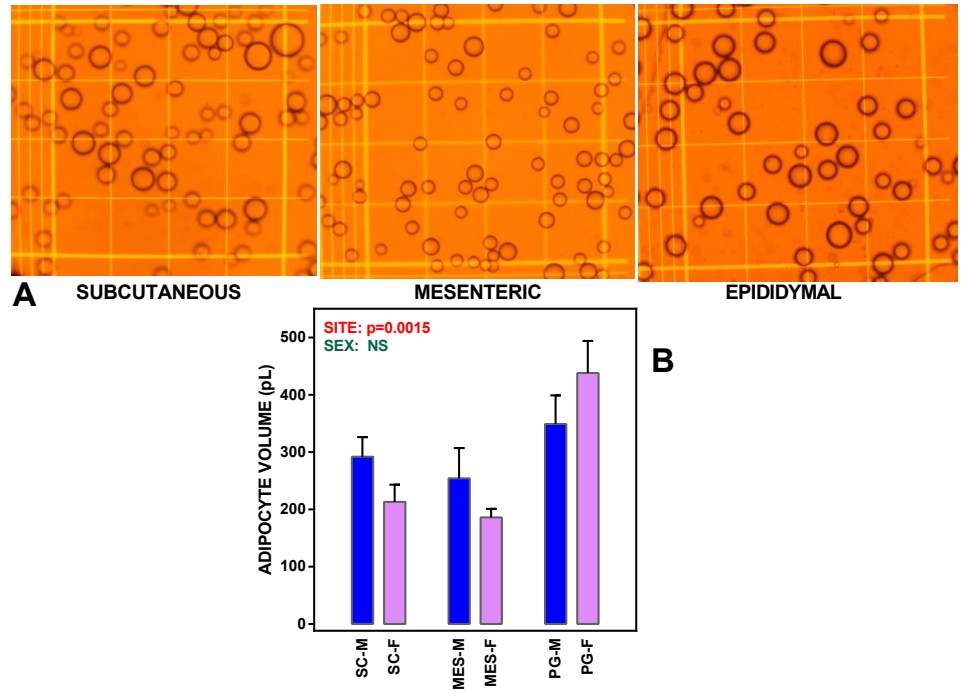

**Figure 1   WAT sites and adipocyte size.** (A) depicts microphotographs of representative isolated adipocyte preparations from subcutaneous, mesenteric and epididymal WAT. The cells were viewed in a Neubauer chamber. Main grid squares were 250 μm wide (and had a volume of 6.25 nL each). (B) shows the mean adipocyte volume in WAT sites extracted from undisturbed female and male adult Wistar rats. The data are the mean ± sem of four groups of two rat-pools for sex and site. SC, subcutaneous WAT; MES, mesenteric WAT; PG perigonadal (epididymal –males, periovaric –females) WAT; blue bars, M, male rats; mauve (clear) bars, F, female rats. Statistical comparison between groups: two-way ANOVA; the $p$ values for sex (green) and site (red) are included in the figure. NS, not significant ($p > 0.05$).

The media were used for the estimation of glucose, lactate, glycerol and NEFA. Glucose was measured using a glucose oxidase-peroxidase kit (#11504, Biosystems, Barcelona, Spain) containing 750 nkat/mL mutarrotase (porcine kidney, 136A5000; Calzyme, St Louis, MO, USA) (*Oliva et al., 2015*). Lactate was measured with kit 1001330 (Spinreact, Sant Esteve d'en Bas, Spain); glycerol was estimated with kit #F6428 (Sigma-Aldrich). NEFA were measured using kit NEFA-HR (Wako Life Sciences, Mountain View, CA, USA).

## Gene expression analyses

Total cell RNA was extracted from the packed washed adipocytes using the Tripure reagent (Roche Applied Science, Indianapolis, IN, USA), and were quantified using a ND-1000 spectrophotometer (Nanodrop Technologies, Wilmington, DE, USA). RNA samples were reverse transcribed using the MMLV reverse transcriptase (Promega, Madison, WI, USA) system and oligo-dT primers. Real-time PCR (RT-PCR) amplification was carried out using 10 μL amplification mixtures containing the Power SYBR Green PCR Master Mix (Applied Biosystems, Foster City, CA, USA), 4 ng of reverse-transcribed RNA and 150
nM primers. Reactions and measurement of evolved fluorescence were analyzed in an ABI PRISM 7900 HT detection system (Applied Biosystems),

A semi-quantitative approach for the estimation of the concentration of specific gene mRNAs per unit of tissue weight was used (*Romero et al., 2007*). *Arbp* was the charge control gene (*Bamias et al., 2013*). The results were expressed as the number of transcript copies per cell in order to obtain comparable data between the groups. The genes analyzed and a list of primers used are presented in Table 1.

## Statistics

The experimental design combined two different animal origins of WAT (female, male), from three different anatomical locations (site: subcutaneous, mesenteric and perigonadal). In addition, the length of incubation (24 h or 48 h) and the initial level of glucose (7 mM or 14 mM). Since the effects of the parameters studied could influence the results in widely different ways, we tried to apply a escalating statistical comparison set up that will allow us to go from a general view to the specifics of which factors were more (or less) responsible for the differences. We considered that the intermixing of data could only be discerned using ANOVA analyses, since our experimental design had low N values, because of the need for integral comparison of data and cooperative analysis in the individual wells. Thus, we scaled the statistical analyses between groups using four-, three- or two-way ANOVAs on the same sets of data. Analyses were carried out with the Stratgraphics program (Statpoint Technologies, Warrington, VA, USA).

# RESULTS

## Cells, glucose uptake and metabolite efflux

Mean adipocyte volumes for the suspensions of cells used are shown in Fig. 1. There were significant differences for site (but not for sex) in mean cell size; the largest cells were those of PG WAT. We expressed the metabolite efflux (or uptake in the case of glucose) as rates akin to the unit of enzyme activity, katal, and taking the cell as unit for comparison. Consequently, the data (shown in Fig. 2) were expressed as attomoles per second and cell. Data were analyzed using ANOVAs, four way, as indicated in the legend of the Figure, three way, independently for each site, and two-way for sex in each site (both series shown in the Figure itself).

Four-way ANOVA showed the general trend: there were differences between sites affecting all parameters; sex also showed significant differences for everything except lactate efflux. On the other side, doubling glucose concentration only affected glucose uptake, whereas the time of incubation affected all parameters except glucose uptake. The latter parameter was closely linked to the incubation time, with limited effects of sex (except in PG WAT).

Lactate efflux was remarkably uniform and not related to glucose concentration or the time of incubation (except in PG WAT), but there was an overall significant effect of sex on SC WAT (not observed in the other sites). Glycerol and NEFA efflux showed a similar pattern, despite glycerol efflux being, in all cases, much larger than that of NEFA. There was a significant effect of sex in all WAT sites (except for NEFA in PG WAT), and a practically

**Table 1  List of primers used in the present study.**

| Gene | Protein | Direction | Sequences | bp |
|---|---|---|---|---|
| (Glut-1) Slc2a1 | Glucose transporter type 1, erythrocyte/brain | 5'> | GCTCGGGTATCGTCAACACG | 97 |
| | | >3' | ATGCCAGCCAGACCAATGAG | |
| Hk1 | Hexokinase type 1 | 5'> | TGGATGGGACGCTCTACAAA | 100 |
| | | >3' | GACAGGAGGAAGGACACGGTA | |
| Pfkl | Phospho-fructokinase, liver, b-type | 5'> | CAGCCACCATCAGCAACAAT | 90 |
| | | >3' | TGCGGTCACAACTCTCCATT | |
| Phgdh | Phospho-glycerate dehydrogenase | 5'> | CTGAACGGGAAGACACTGGGAA | 138 |
| | | >3' | AACACCAAAGGAGGCAGCGA | |
| G6pdx | Glucose-6-phosphate dehydrogenase X-linked | 5'> | GACTGTGGGCAAGCTCCTCAA | 77 |
| | | >3' | GCTAGTGTGGCTATGGGCAGGT | |
| Me1 | NADP$^+$-dependent malic enzyme | 5'> | GGAGTTGCTCTTGGGGTAGTGG | 143 |
| | | >3' | CGGATGGTGTTCAAAGGAGGA | |
| Ldha | L-lactate dehydrogenase a | 5'> | AAAGGCTGGGAGTTCATCCA | 96 |
| | | >3' | CGGCGACATTCACACCACT | |
| Ldhb | L-lactate dehydrogenase b | 5'> | GCGAGAACTGGAAGGAGGTG | 145 |
| | | >3' | GGGTGAATCCGAGAGAGGTTT | |
| Pdk4 | Pyruvate dehydrogenase kinase, isoenzyme 4 | 5'> | CTGCTCCAACGCCTGTGAT | 142 |
| | | >3' | GCATCTGTCCCATAGCCTGA | |
| Pck1 | Phosphoenol-pyruvate carboxykinase, cytosolic | 5'> | CGGGTGGAAAGTTGAATGTG | 142 |
| | | >3' | AATGGCGTTCGGATTTGTCT | |
| Mct1 | Monocarboxylate transporter | 5'> | CCCAGAGGTTCTCCAGTGCT | 133 |
| | | >3' | ACGCCACAAGCCCAGTATGT | |
| Cpt1b | Carnitine O-palmitoleoyl-transferase 1, muscle isoform | 5'> | TGCTTGACGGATGTGGTTCC | 152 |
| | | >3' | GTGCTGGAGGTGGCTTTGGT | |
| Gpd1 | Glycerol 3-phosphate dehydrogenase | 5'> | CTGGAGAAAGAGATGCTGAACG | 113 |
| | | >3' | GCGGTGAACAAGGGAAACTT | |
| Gpam | Glycerol-3-phosphate acyltransferase, mitochondrial | 5'> | GGTGAGGAGCAGCGTGATT | 129 |
| | | >3' | GTGGACAAAGATGGCAGCAG | |
| Pgp | Phosphoglycolate phosphatase | 5'> | CCTGGACACAGACATCCTCCT | 100 |
| | | >3' | TTCCTGATTGCTCTTCACATCC | |
| Gk | Glycerol kinase | 5'> | ACTTGGCAGAGACAAACCTGTG | 74 |
| | | >3' | ACCAGCGGATTACAGCACCA | |
| Aqp7 | Aquaporin 7 | 5'> | ACAGGTCCCAAATCCACTGC | 127 |
| | | >3' | CCGTGATGGCGAAGATACAC | |
| CD36 | Platelet glycoprotein 4 [fatty acid transporter] | 5'> | TGGTCCCAGTCTCATTTAGCC | 154 |
| | | >3' | TTGGATGTGGAACCCATAACTG | |
| Acaca | Acetyl-CoA carboxylase 1 | 5'> | AGGAAGATGGTGTCCGCTCTG | 145 |
| | | >3' | GGGGAGATGTGCTGGGTCAT | |
| Fas | Fatty acid synthase | 5'> | CCCGTTGGAGGTGTCTTCA | 117 |
| | | >3' | AAGGTTCAGGGTGCCATTGT | |

**Table 1** (*continued*)

| Gene | Protein | Direction | Sequences | bp |
|------|---------|-----------|-----------|-----|
| *Acly* | ATP citrate lyase | 5'> | TGTGCTGGGAAGGAGTATGG | 137 |
| | | >3' | GCTGCTGGCTCGGTTACAT | |
| *Lpl* | Lipoprotein lipase | 5'> | TGGCGTGGCAGGAAGTCT | 116 |
| | | >3' | CCGCATCATCAGGAGAAAGG | |
| *(Hsl)* *Lipe* | Lipase, hormone sensitive | 5'> | TCCTCTGCTTCTCCCTCTCG | 108 |
| | | >3' | ATGGTCCTCCGTCTCTGTCC | |
| *(Atgl)* *Pnpla2* | Adipose triacylglycerol lipase | 5'> | CACCAACACCAGCATCCAAT | 120 |
| | | >3' | CGAAGTCCATCTCGGTAGCC | |
| *Arbp* | 0S acidic ribosomal phospho-protein PO [housekeeping gene] | 5'> | CCTTCTCCTTCGGGCTGAT | 122 |
| | | >3' | CACATTGCGGACACCCTCTA | |

nil effect of glucose concentration. However, the duration of incubation markedly raised the efflux of both glycerol and NEFA in all sites, irrespective of sex, the sole exception being adipocytes of MES WAT from females, which were unaffected by glucose and/or incubation time. Glycerol efflux (and NEFA to a lower extent) by MES cells was higher than those of SC or PG cells.

## Analysis of gene expression

We used the same statistical comparative approach described above for metabolites to analyze the changes in gene expression at 24 and 48 h of incubation. Here we present the data as number of copies of the gene mRNA transcripts per cell. Figures 3 to 6 show the gene expressions of main enzymes and transporters affecting the metabolism of glucose, lipogenesis and 3C fragment handling in the adipocyte. The statistical analysis of the differences between groups, related to WAT sites, sexes and the medium glucose concentration were shown in the graphs and Figure legends. The presentation—and initial analysis—of the gene expression data was done along three main lines: (a) glucose and glycerol, (b) fatty acid metabolism, (c) pyruvate, lactate and oxaloacetate. Figure 7 shows a schematic view of the genes studied. superimposed to the main metabolic pathways of carbohydrate-lipid relationships in the adipocyte.

### (a) Glucose and glycerol

In MES, the expression of *Glut1* was higher in males than in females; it was practically unaltered by glucose and incubation time. In SC WAT, the *Glut1* expression was lower than in the other sites. Males showed a marked interaction between glucose level and incubation time. The expression of *Hk1* repeated the trend of *Glut1*, but in MES WAT, the effects of sex were more marked, with male-origin adipocytes decreasing their gene expression with incubation time. Sex differences were more marked in *Pfkl*, repeating the pattern of higher male expression in MES and lower in PG WAT; no effect of glucose levels were observed, either, but incubation time increased the expression of this gene, especially in PG WAT. The expressions of *Phgdh* (not a pathway control enzyme) were rather uniform, and lower than the other glycolytic enzymes analyzed. There was a trend to increase expression with incubation time, and to maintain the male–female predominance described for the other

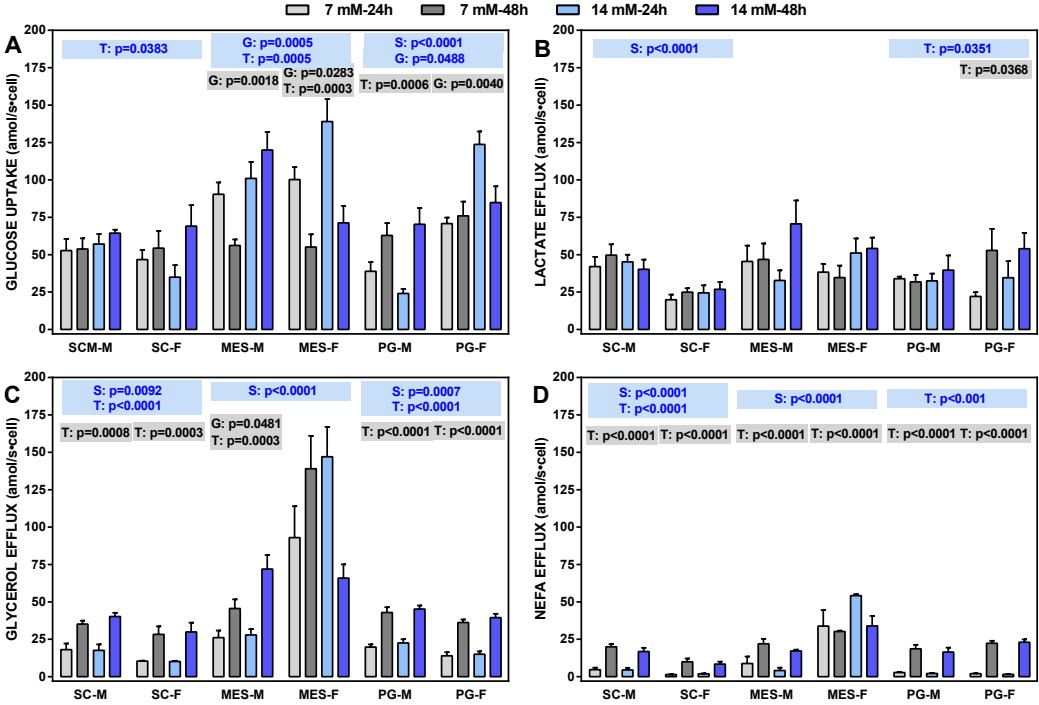

**Figure 2 Glucose uptake rates and lactate, glycerol and NEFA efflux rates (all in attomoles per second and cell) of adipocytes obtained from three WAT sites of male and female adult Wistar rats.** The data are the mean ± sem of four groups of two rat-pools for sex and site, incubated for 24 or 48 h. The abbreviations for site and sex are the same as in Fig. 1. (A) Glucose uptake, (B) lactate effl, (C) glycerol efflux, (D) NEFA efflux. The statistical significance of the differences between groups was investigated using two-, three- and four-way ANOVA analyses and the variables: WAT Site, Sex (S), medium Glucose concentration (G) and Time of incubation (T). The figure shows the three- (S, G, T) and two-way ANOVA (G,T) analyses for each site; the data are included in blue boxes (three-way) encompassing each site and showing the corresponding $p$ values when statistically significant ($p < 0.05$), or gray boxes (two-way) with significant $p$ values for each sex within a given WAT site. The four-way ANOVA values were, Site: $p < 0.001$ for all parameters except $p = 0.0081$ (lactate): Sex: $p < 0.0001$ for glycerol and NEFA, and $p = 0.0046$ for glucose; Glucose concentration: $p < 0.0001$ for glucose; Time of incubation: $p < 0.0001$ for NEFA, $p = 0.0001$ for glycerol and $p = 0.0133$ for lactate.

enzymes, but variability was high and the statistical significance of the differences was low. In sum, no marked changes were found to be influenced by the conditions of the study, suggesting a fluid and uniform operation of glycolysis down to pyruvate with practically no effects of external glucose concentration, combined with increased expressions with incubation time.

Glycerol efflux was paralleled by a marked trend to increase the already high expression of *Gpd1* (i.e., compared with *Phgdh*) with incubation time, but –again– it was not affected by the concentration of glucose. The maximal increase in expression, not linked to sex, was observed in PG WAT. The phosphatase pathway (*Pgp*) showed no changes at all, except for higher values in females of its (also high) expression in PG WAT. This was not the case for the enzyme catalyzing the reverse reaction, glycerokinase, which gene (*Gk*) maintained the differences between sexes, but showed a higher increase of its expression

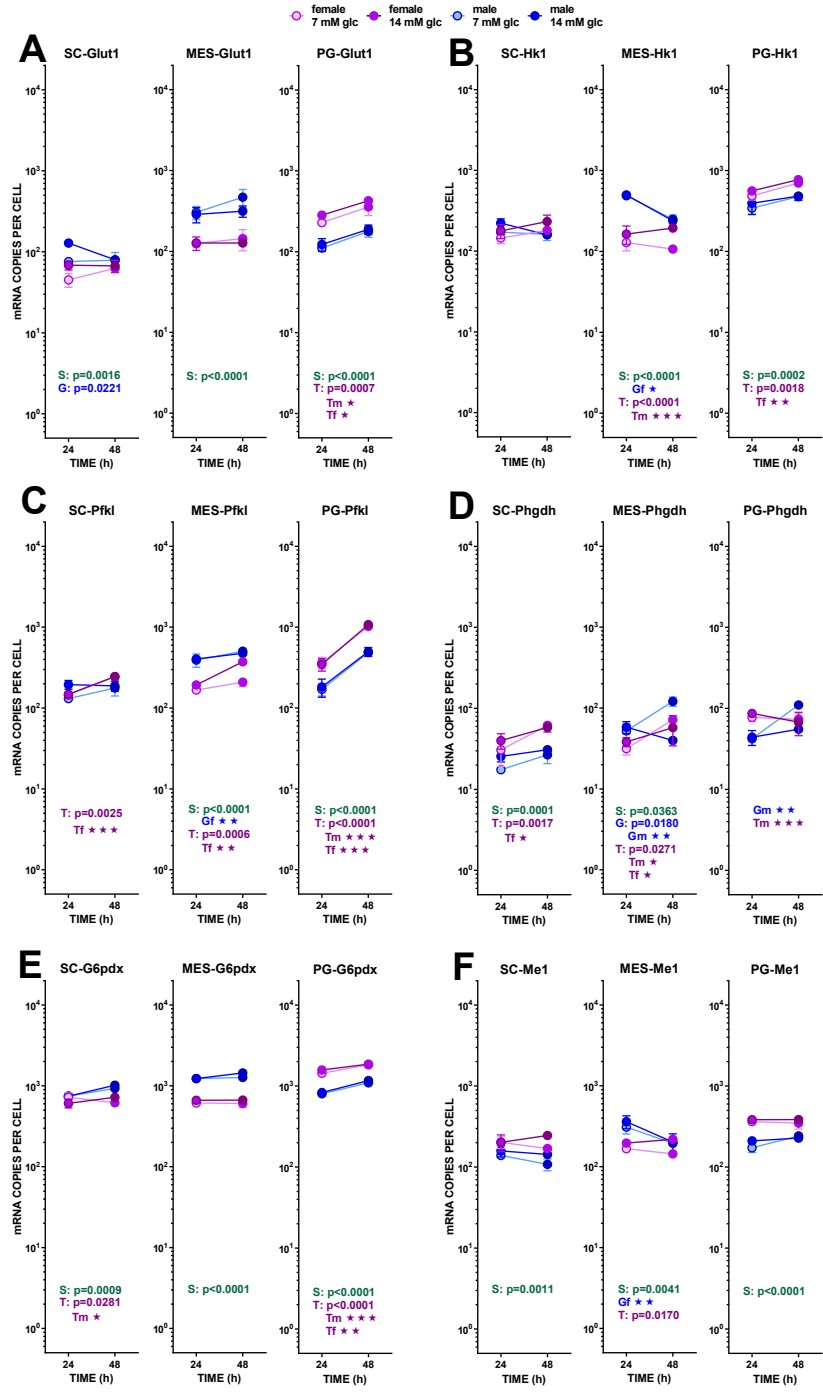

**Figure 3** **Expression of genes related with the metabolism of glucose, lipogenesis and 3C handling in adipocytes of male and female adult rats incubated in the presence of 7 mM or 14 mM glucose (I).** The data are the mean ± sem of four (or three) groups of two rat-pools for sex and site, incubated for 24 or 48 h, and represent the number of mRNA copies per cell of each gene. The data are presented on a log scale. Blue circles indicate males: deep blue 14 mM glucose, 

**Figure 3 (…continued)**
light blue 7 mM glucose; purple circles represent females: violet-purple 14 mM glucose, mauve 7 mM glucose. SC, subcutaneous, MES, mesenteric and PG, perigonadal WAT. (A) *Glut1*, (B) *Hk1*, (C) *Pfkl*. (D) *Phgdh*, (E) *G6pdx*, (F) *Me1*. Statistical analysis of the differences between groups: the figures contain the $p$ values for a three-way ANOVA analysis (sex (S in green), glucose (G in blue) and time of incubation (T in purple)) applied to each site. Results of two-way ANOVA for these paramers are also represented: the letters and colors are the same, adding m for male and f for female subgroups; for two-way ANOVAs, the $p$ values are represented by up to three stars, corresponding to three levels of statistical significance of the differences ($p < 0.05$, $p < 0.01$ and $p < 0.001$). Only significant differences have been represented. The results for the four-way ANOVAs were significant ($p < 0.0001$) for time in all gene expressions. The effect of sex was significant only for *Pfkl* ($p = 0.0006$) and *Me1* ($p = 0.0007$), that of glucose concentration was significant for *Hk1* ($p = 0.0371$), *Phgdh* ($p = 0.0164$) and *Me1* ($p = 0.206$). The effect of time of incubation was significant for *Glut1* ($p = 0.0023$) and *Pfkl*, *Phgdh*, and *G6pdx* ($p < 0.0001$). Non-statistically significant data ($p > 0.05$) were not represented.

with incubation time. The overall number of copies per cell of *Gk* was, however, about one order of magnitude lower than that of *Pgp*. The patterns for Aquaporin 7 gene (*Aqp7*) expression closely resembled those of *Pgp*, including the range of copies per cell.

The assumed incorporation of newly formed glycerol-3P into acyl-glycerols increased with incubation time, judging from the expression of *Gpam*, following a pattern comparable to that of *Gpd1*, deeply affected by time but not by medium glucose levels. Sex differences were maintained: higher values for females in SC, and, especially, PG WAT. and higher levels of expression for males in MES.

### (b) Fatty acid metabolism

Lipogenesis did not seem to represent a quantitatively important process under the conditions tested because of the limited possibility of producing acetyl-CoA from the main medium substrate, glucose (down to pyruvate). Pyruvate dehydrogenase activity may be hampered, both because of the relatively scarce number of mitochondria and because of the proportionally high expression of the main controller of the enzyme, pyruvate dehydrogenase kinase 4. The expression of its gene (*Pdk4*) showed a marked effect of incubation time, increasing (in all three sites) differently according to sex. Males and females' *Pdk4* expression in SC and PG WAT increased about one order of magnitude in 24 h. In MES WAT, males followed the same pattern, but no significant increase with incubation time was observed in females (despite its spectacular rise in PG WAT). Under these conditions, pyruvate dehydrogenase could not operate fully at 48 h, but these effects should be less marked in the MES WAT of females.

The transfer of acetyl-CoA to the cytoplasm via citrate: ATP lyase was not affected by the treatment received by adipocytes, as shown by the little change found in the expression of its gene *Acly*. Carboxylation of acetyl-CoA to malonyl-CoA was probably unchanged (or decreased in male adipocytes of MES WAT) from the expression of *Acaca*. A similar pattern, with higher number of copies per cell, was observed in *Fas*, fatty acid synthase. As for the availability of NADPH in the cytoplasm, the indicator gene of the pentose phosphate pathway *G6pdx* showed high numbers of copies and a marked sexual differentiation in MES and PG WAT, but no changes induced by glucose or incubation time. These data suggest an absence of change in the main NADPH provider, in parallel to the glycolytic

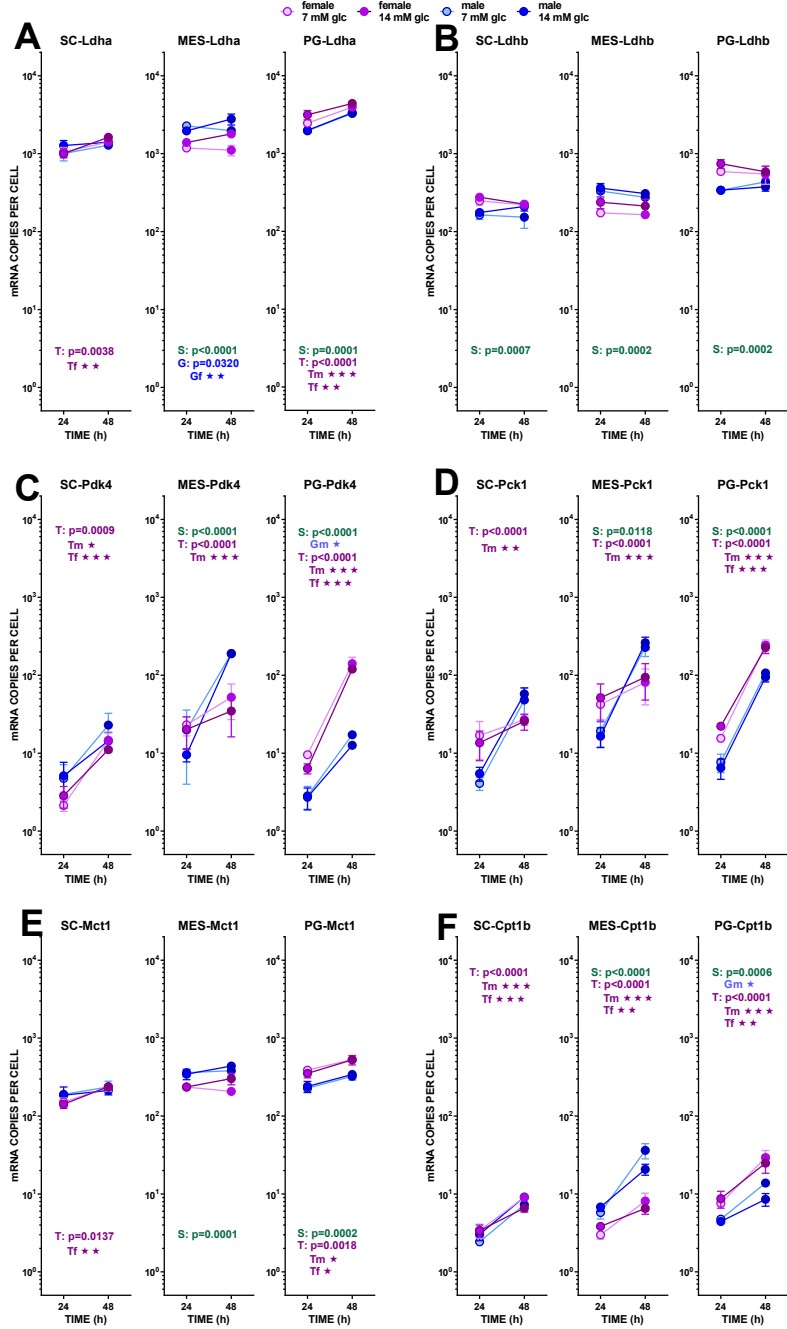

**Figure 4 Expression of genes related with the metabolism of glucose, lipogenesis and 3C handling in adipocytes of male and female adult rats incubated in the presence of 7 mM or 14 mM glucose (II).** The data are the mean ± sem of four (or three) groups of two rat-pools for sex and site, incubated for 24 or 48 h, and represent the number of mRNA copies per cell of each gene. The distribution figure set-up and other conventions are those described in Fig. 4. (1) *Ldha*, (B) *Ldhb*, (C) *Pdk4*, (D) *Pck1*, (E) *Mct1*, (F) *Cpt1b*. The results for the four-way ANOVAs were significant ($p < 0.0001$) for time in all gene expressions. The effect of sex was significant only for *Ldhb* ($p = 0.0063$), that of glucose concentration was significant only for *Ldha* ($p = 0.0037$). The effect of time of incubation was significant ($p < 0.0001$) for all genes except *Ldhb* (NS).

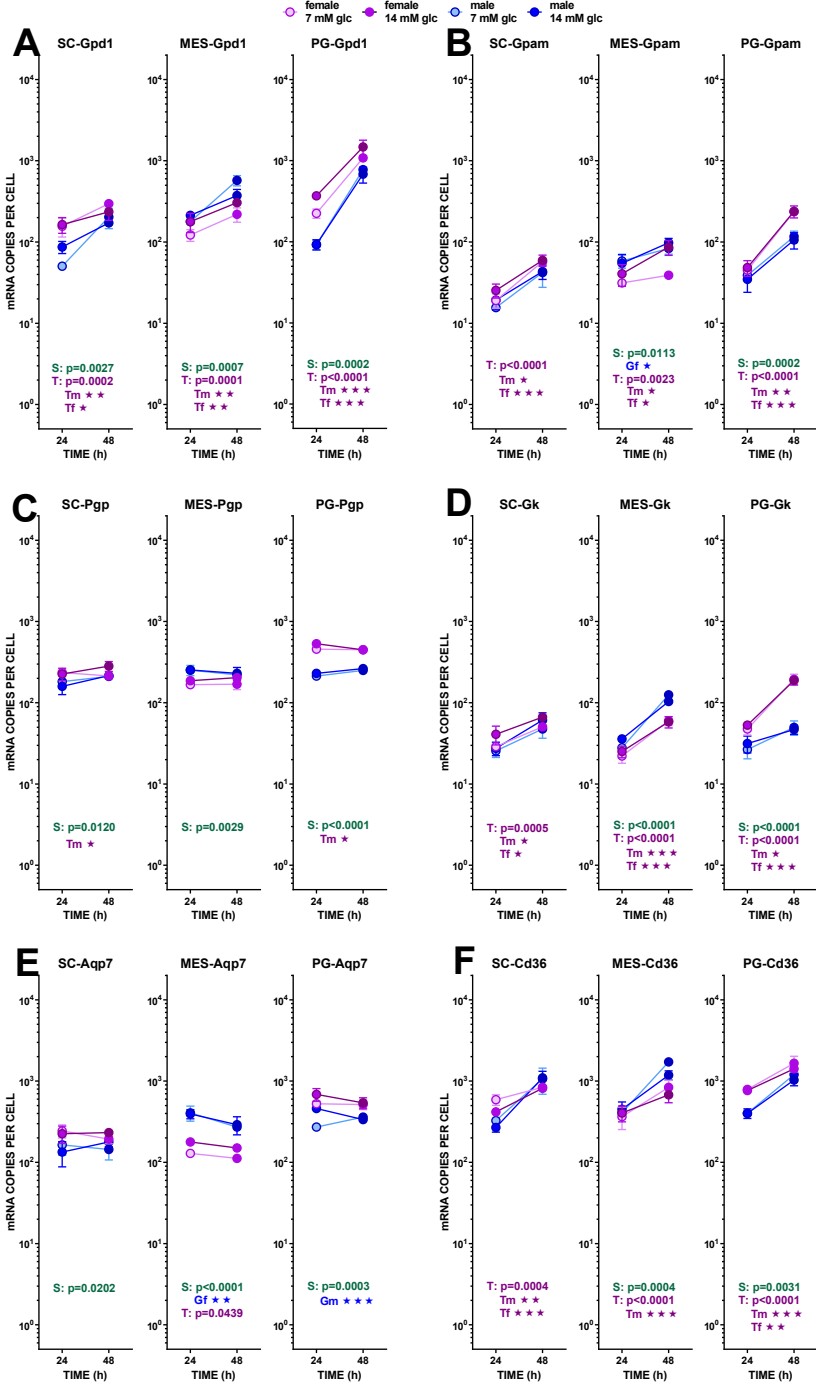

**Figure 5 Expression of genes related with the metabolism of glucose, lipogenesis and 3C handling in adipocytes of male and female adult rats incubated in the presence of 7 mM or 14 mM glucose (III).** The data are the mean ± sem of four (or three) groups of two rat-pools for sex and site, incubated for 24 or 48 h, and represent the number of mRNA copies per cell of each gene. (continued on next page...)

and pyruvate-handling pathways already analyzed. The other main NADPH generator, malic enzyme (*Me1*), also showed little change in its expression, with a tendency to lower the number of copies in MES-WAT with incubation time.

Probably, the uptake of medium fatty acids (at least from cell remnants), was activated, since the expression of one of the main transporters *Cd36* increased with time. Sex affected only PG WAT, and no effects of glucose levels were observed at all. Mitochondrial utilization of acyl-CoA was assumed to be low (if any), first because of the nature of the tissue and its energy needs (expected low oxidative metabolism); second because of the ample availability of glucose; and third because the expression of *Cpt1b* was very low in all three sites, thus making difficult the entry of acyl-CoA into the mitochondria. The already described differences related to sex were maintained, and there was a marked increase with time in the expression of this gene in both sexes.

The main lipases of adipose tissue: the extracellular lipoprotein lipase (*Lpl*), and the internal adipose TAG lipase (*Atgl*) and hormone-sensitive lipase (*Hsl*) showed essentially the same trend. All had uniform increases in gene expression with incubation time, lack of effects of glucose and sex-related differences of limited extent. *Lpl* showed the highest number of copies per cell found in this study. In the case of MES WAT, females' increase of expression with time was less marked than that of males, a reminiscence of the discordance described for *Pdk4*. The high coordinated increase in lipase activity was not correlated to the limited release of NEFA into the medium, and these levels were far from being correlated with any of the expressions of the lipases studied.

### (c) Pyruvate, lactate and oxaloacetate

In addition to the different sex-related response of *Pdk4* expression of adipocytes with incubation, further limiting the synthesis of acetyl-CoA in most of the conditions analyzed, the obviously major outlet for excess pyruvate generation in the cytoplasm was its conversion to lactate, completing the classic glycolytic pathway. The number of copies of *Ldha* was high, with practically no effect of sex, but showing a trend to increase its expression with time of incubation in SC and PG WAT. Initial glucose affected more intensely the MES values, decreasing *Ldha* expression at lower-, and increasing at higher medium glucose, with a clear differentiation by sex (higher values for males). *Ldhb* showed less change and lower number of copies per cell. The ratio of expression of both lactate dehydrogenase isoform genes (*Ldha/Ldhb*) was rather constant, with a mean value of 6.2 for both sexes. The expression of the monocarboxylate transporter gene (*Mct1*) presented a pattern closely similar to that of *Ldha*, hinting at a coordinated regulation. As described above, *Me1* showed only limited changes in expression, which agrees with the lower needs for NADPH (mainly used for lipogenesis) observed under the conditions of incubation.

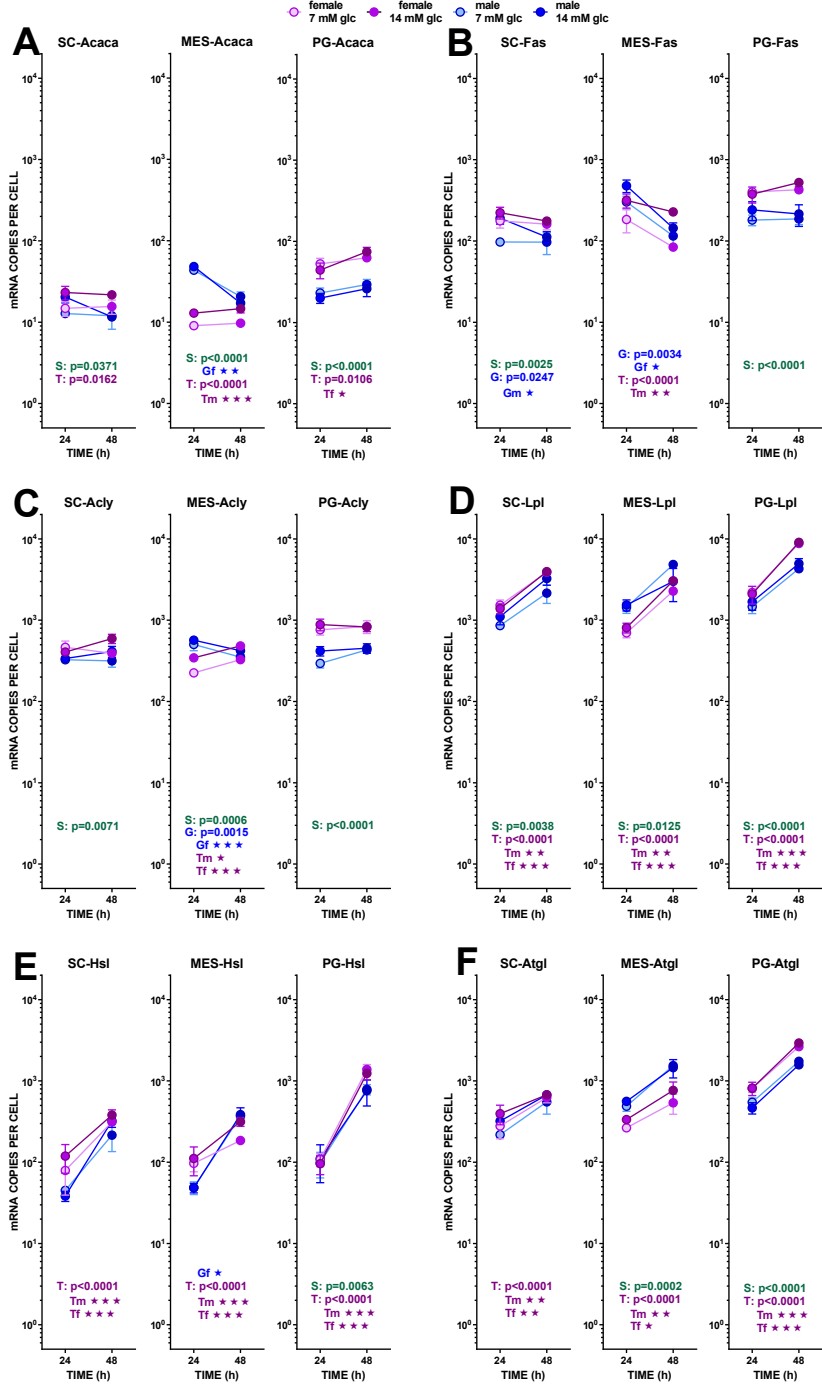

**Figure 6 Expression of genes related with the metabolism of glucose, lipogenesis and 3C handling in adipocytes of male and female adult rats incubated in the presence of 7 mM or 14 mM glucose (IV).** The data are the mean ± sem of four (or three) groups of two rat-pools for sex and site, incubated for 24 or 48 h, and represent the number of mRNA copies per cell of each gene. (continued on next page...)

**Figure 6 (…continued)**
The distribution figure set-up and other conventions are those described in Fig. 4. (1) *Acaca*, (B) *Fas*, (C) *Acly*, (D) *Lpl*, (E) *Hsl*, (F) *Atgl*. The results for the four-way ANOVAs were significant ($p < 0.0001$) for time in all gene expressions. The effect of sex was significant for *Fas* and *Acly* ($p < 0.0001$), *Lpl* ($p = 0.0001$, *Acaca* ($p = 0.0020$) and *Hsl* ($p = 0043$); that of glucose concentration was significant only for *Fas* ($p = 0.0003$) and *Acly* ($p = 0.0118$). The effect of time of incubation was significant for *Lpl*, *Hsl*, *Atgl* ($p < 0.0001$), and *Fas* ($p = 0–014$).

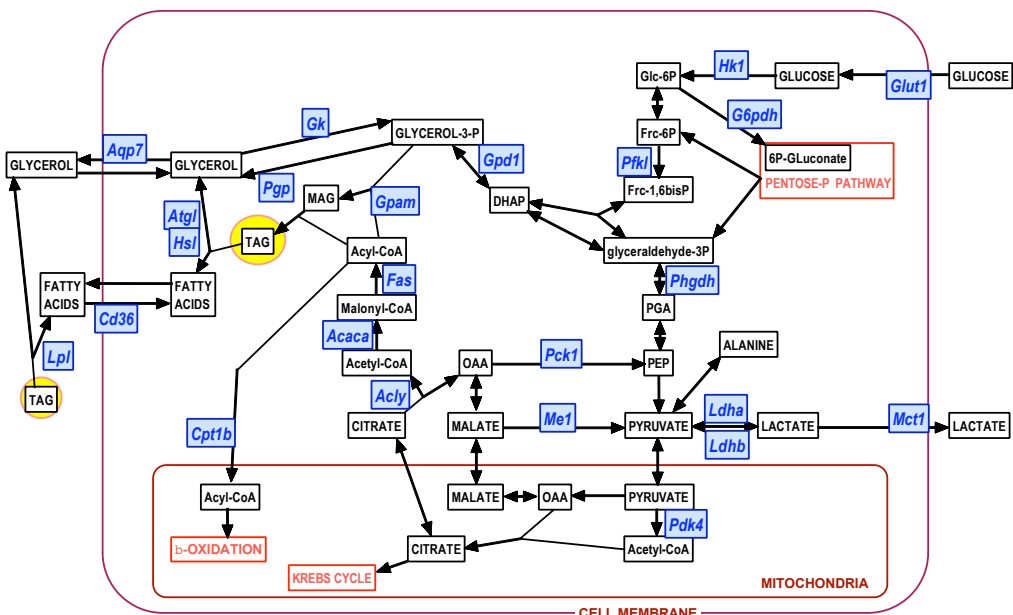

**Figure 7** **Diagram showing the position of the genes investigated on a scheme of the main adipocyte energy metabolism pathways.** Blue: genes (for enzymes and transporters); white: metabolites.

Last, but not least, the key marker enzyme gene *Pck1* (P-enol-pyruvate carboxy-kinase), a critical enzyme bridging the pyruvate kinase gap to favor the arrival of oxaloacetate to the triose-P pool under conditions of scarcity of glucose, showed a marked sex-related difference in its response to incubation. Irrespective of glucose concentration, both sexes in PG and only males in SC and MES WAT increased dramatically its expression with time of incubation, suggesting the need for the conversion of excess cytoplasmic oxaloacetate into P-enol pyruvate. This change was not observed in female adipocytes of SC and MES WAT, in a pattern closely resembling that of MES *Pdk4*.

## DISCUSSION

The main outcome of this study is a reinforcement of the widely accepted idea that WAT sites are remarkably uniform (*Arriarán et al., 2015b*) from a qualitative point of view, but also that they are adjusted to play different metabolic functions depending on their location (*Prunet-Marcassus et al., 2006*). Thus, the differences between sites were mainly quantitative. This idea is not new, but the comparative analyses of the capacity of metabolic

pathways are too scarce and incomplete to provide sufficient support. Our contribution may help reinforce the assumption that WAT plays a role in glucose handling, sufficient to participate in the maintenance of glycaemia (*Arriarán et al., 2015b*; *Ho-Palma et al., 2016*). We also present additional evidence that key metabolic functions of WAT are deeply affected by sex, as previously indicated (*Amengual-Cladera et al., 2012*; *White & Tchoukalova, 2014*). This aspect is more clearly observed in MES and SC WAT than in the fat pads (PG) attached to reproductive organs. Our data also agree with the postulated role of glycolysis to lactate as main provider of energy (ATP) to sustain most of the white adipocyte normal function (*Crandall et al., 1983*; *Hagström et al., 1990*; *Sabater et al., 2014*), thus reducing its need for oxidative metabolism, and, consequently, of oxygen (*Sjöström, 1990*). This aspect is also sustained, by the WAT relative scarcity of mitochondria and, logically, of the oxidative metabolism they sustain (*Deveaud et al., 2004*; *Frayn, Langin & Karpe, 2008*). We assume that glucose is taken up as needed (mainly for energy); and, consequently, lactate is released not as a response to glucose availability (and uptake) but depending on cytoplasm energy requirements. The consequence is an extensive conversion of glucose to 3C metabolites because of the energetic inefficiency of the (anaerobic) glycolytic use of glucose. The 3C thus generated can be easily used for energy or as carbon source elsewhere (*Frayn & Coppack, 1990*; *Sabater et al., 2014*; *Arriarán et al., 2015c*; *Rotondo et al., 2017*).

Our previous studies on adipocyte primary cultures were carried out using only male rats' epididymal adipose tissue (*Rotondo et al., 2017*), a classical depot WAT site; the comparison done here with female rat periovaric WAT showed little differences between them, even with respect to potential fatty acid synthesis. These results were not expected, given the higher tendency to fat accumulation of adult males versus females, both in rats and humans (*Schemmel, Mickelsen & Tolgay, 1969*; *Fried, Lee & Karastergiou, 2015*). However, as a rule, the number of copies per cell for most of the genes studied was higher in females than in males, in contrast with the smaller (NS) mean size of their adipocytes, hinting at a possibly higher overall metabolic activity of female WAT (*Blaak, 2001*).

MES WAT showed a markedly higher efflux of glycerol, and NEFA (to a proportionally lower extent in molar terms), compared with the other sites investigated. Previous analyses of glucose carbon fate in epididymal WAT showed that a sizeable proportion of glucose ended forming part of glyceride-glycerol (*Cahill, Leboeuf & Renold, 1959*; *Ho-Palma et al., 2016*). Results were more marked after incubation of adipocytes for more than one day in the presence of glucose (*Rotondo et al., 2017*). There was, also, a significant lipase-driven TAG turnover, in which most of fatty acids were recycled to TAG (*Hammond & Johnston, 1987*), but glycerol was excreted (*Vaughan, 1962*; *Jansson et al., 1992*). The data of the present study agree with this interpretation. However, since no labelled C has been used, we can only deduce (not prove) the origin of glyceride-glycerol in the adipocytes. We know, however, that the rates of glycerol release to the medium by MES were much higher than in PG WAT (both epididymal and periovaric). We can assume, thus, that glycerol production in MES WAT may be higher than that of the only tissue quantitatively analyzed in detail with tracers, male PG WAT (*Ho-Palma et al., 2016*). Since MES WAT plays an ancillary energy-handling role to the liver, its massive production of glycerol (to our knowledge not previously described) may help facilitate the hepatic handling of fatty acids. Intestinal

absorption-derived fatty acids, not used in the synthesis of the TAG carried out by lymph, as well as excess systemic blood NEFA arrived to the liver mixed in the portal blood with the efflux of MES-WAT. This WAT site may also help to lower the portal intestine-released glucose load carried to the liver, as shown by the higher glucose uptake and 3C substrate efflux rates in MES when compared with SC and, especially, PG WAT. We can speculate that MES-WAT activity may help buffer the impact of large digestive glucose loads, and thus facilitate its hepatic handling.

The differences in response between female- and male-derived MES adipocytes facing periodic exposure to excess energy (in this case, glucose) seem minimal, but may be far-reaching. In addition to higher overall glycerol efflux, in female rats, pyruvate dehydrogenase inhibition by its kinase 4 seems not to be altered by either glucose levels or time of incubation. This occurs in contrast with the high increases elicited in males (*Rotondo et al., 2017*) of the number of copies of *Pdk4*, a powerful inhibitor of the dehydrogenase (*Cadoudal et al., 2008*), which is mainly regulated via transcription (*Jeong et al., 2012*). This increase was also observed in SC and PG WAT of both male and female rats, being, thus, a unique effect restricted to (female) MES WAT. This assumed "lower poential inhibition" of pyruvate dehydrogenase hints at a potentially higher flow of 3C (pyruvate) into mitochondrial acetyl-CoA, thus facilitating either its oxidation or incorporation into the lipogenic pathway. This could not proceed so easily in males (or in other WAT sites of females), which *Pdk4* expression increased with time and exposure to glucose, preventing the decarboxylation of mitochondrial pyruvate to acetyl-CoA. Since in female MES, excess mitochondrial pyruvate could not be processed to acetyl-CoA, it must be returned to the cytosol. The limitation of mitochondrial acetyl-CoA means that it could not be derived through the Krebs cycle, which in any case needs oxygen to oxidize it. Thus, the most probable way of utilization of the 'surplus' mitochondrial pyruvate is via carboxylation (*Ballard & Hanson, 1967*) to oxaloacetate and its transfer to the cytoplasm via the pyruvate/malate shuttle, partly using the machinery of fatty acid synthesis (*Patel et al., 1971*). This implies a considerable activity of the malate-oxaloacetate metabolism in the mitochondria, albeit not implicating citrate. After malate is transferred into the cytosol, it could be either used by the malic enzyme to provide NADPH (improbable in this case as explained above) or oxidized to oxaloacetate by the cytoplasmic malate dehydrogenase, providing NADH (*Nye et al., 2008*). Cytoplasmic oxaloacetate can be converted into P-enol-pyruvate by its carboxy-kinase (*Ballard, Hanson & Leveille, 1967*). In males, the expression of its gene, *Pck1*, was raised with incubation time in parallel to *Pdk4*, but it was stabilized in females following the same pattern than the kinase, an effect extended to SC-WAT. The differences in the cytosol-mitochondria handling of pyruvate suggest a deep sex-related divergence in the metabolic fate of pyruvate which, ultimately, may help explain the known different metabolic handling of lipids (and probably glucose) by visceral WAT depending on sex (*Franckhauser et al., 2002*).

Curiously, the alternatives for disposal of the oxaloacetate assumedly extracted from the mitochondria by the malate shunt point to the regeneration of P-enol pyruvate. This high-energy compound could not go further up the glycolytic pathway because of massive thrust of glycolysis carbon flow towards pyruvate/lactate. The high lactate efflux is proof of

the unequivocal direction of glycolysis in the adipocytes, at least under the conditions and cell sizes used in this study, leaving open only the conversion (again) of P-enol-pyruvate to pyruvate by pyruvate kinase. However, the difference (in females) lies, precisely, in the assumed provision of cytoplasmic NADH by malate dehydrogenase, which may provide reducing power for the final conversion of pyruvate to lactate (*Rotondo et al., 2017*) easing its disposal via excretion.

Since pyruvate is a good substrate for lipogenesis (*Schmidt & Katz, 1969*) and WAT is the largest depot for body lipid storage, one can expect lipogenesis to be fully activated when glucose availability is high. The process requires, however, the massive production of acetyl-CoA, its transfer to the cytoplasm via citrate and then a full activation of lipogenesis, including necessarily higher expressions of *Acaca* and *Fas* and the activation of NADPH providers (such as *G6pdx* or *Me1*). None of these signs was detected; no effects of glucose or duration of the incubation affected the expressions of these genes. Probably, quantitative conversion of glucose to fatty acids under high glucose conditions was not as a function as relevant for the adipocyte (or WAT) as to dispose of excess glucose, in agreement with previous studies using labelled glucose (*Rotondo et al., 2017*).

In addition to the paradoxical apparent inactivity of lipogenesis, there was a high increment with time (but not with higher glucose) of lipase gene *Lpl* and *Atgl* expressions (in *Hsl* we observed a tendency to "restraint" resembling those of *Pdk4* and *Pck1*) (*Sabater et al., 2014*; *Rotondo et al., 2016*; *Rotondo et al., 2017*). However, the expected massive efflux of NEFA did not occur. In any case, NEFA were released to the medium in much smaller proportions than the canonical three-to-one molar ratio *vs.* glycerol expected from TAG hydrolysis, in disagreement with the frank liberation of NEFA described for the initial phase of catecholamine-elicited WAT lipolysis (*Jocken et al., 2008*). The results shown here were, however, fully compatible with an activation of TAG turnover, a critical adipocyte regulatory system (*Arner et al., 2011*). In the end, this process, selectively released glycerol as a 3C substrate ultimately derived from glucose, as we have previously established (*Arriarán et al., 2015c*; *Rotondo et al., 2017*). The highest number of lipase gene mRNA copies per cell, and the steepest increase with time was observed in PG WAT (*Rotondo et al., 2016*; *Rotondo et al., 2017*), with values even higher for females as found in the present study. Nevertheless, the glycerol release rates observed for this site were smaller than in MES and SC WAT.

The higher female number of copies for *Gk* (glycerol kinase gene) may suggest the existence of an additional restrain of the whole process, i.e., free glycerol being recycled to *sn*-glycerol-3P (*Lee et al., 2005*). This process may potentially decrease the actual release of glycerol, in parallel to lower NEFA liberation. It can unbalance, also, the equilibrium between glycerol-3P synthesis and hydrolysis (*Margolis & Vaughan, 1962*). However, we have not sufficient data to prove or disprove the operation of this alternative pathway, which would help define the postulated increase in WAT turnover as a 'futile cycle' in which energy was wasted (for thermogenesis?).

WAT production of glycerol has been known for a long time (*Vaughan, 1962*; *Van der Merwe et al., 1998*; *Bolinder et al., 2000*). Glycerol has been related to glucose in WAT as source of carbon (*Wood, Leboeuf & Cahill, 1960*; *Smith, 1972*), but its immediate origin

has been linked to lipolysis (*Karpe et al., 2005*; *Langin, 2006*) within the context of the glucose-fatty acid cycle (*Randle et al., 1963*). However, glycerol and fatty acid effluxes seldom have been analyzed together (*Jocken et al., 2008*). The quantitative approach used here shows that there is a direct relationship between glycerol and NEFA release, but not at the ratios expected, and points to lipolysis as part of TAG turnover. This turnover was accelerated with time, apparently to favor the release of more glycerol. Nevertheless, the main factor affected by these changes, and that most modulated by sex (at least in MES adipocytes) was assumed to be the equilibrium between NADPH and NADH in the thin layer of adipocyte cytoplasm. Intercellular cooperation, perhaps close neighboring cells, smaller than adipocytes, providers of mitochondria oxidative power, are needed to fully understand how WAT works, and, also, to uncover the intricate effects of sex on WAT operation (*Randle et al., 1963*).

## CONCLUSIONS

The fairly uniform (or coordinated) pattern of gene expressions, with limited overall changes, and the actually small amount of "live matter" in the adipocytes used in the incubations hint at an active but stabilized energy sustaining machinery working along the 2-day incubation. This resulted in a high proportion of glucose converted to 3C substrates, largely lactate. The length of the incubation period modulated the development and timing of the alternative mechanisms that allow the synthesis and release of free glycerol, especially addressed to the liver (at least by the highest WAT glycerol producer, the MES site), and the considerable restraint observed in the synthesis of fatty acids in the presence of high medium glucose concentrations. These findings are not completely new, but its concatenation is.

In fact, the resilience of adipocytes to limit the production of additional fatty acids under excess glucose availability—instead breaking up most of the glucose, which was then released as lactate and glycerol—is remarkable. It also makes us wonder whether we can keep considering true the current assumption that WAT 'has to take up and use for fat synthesis the surplus dietary glucose', under conditions of excess glucose/energy available (a principle often used to help explain obesity).

### Funding

The authors received no funding for this work. Part of the expenses have been endorsed by the researchers themselves and by the University of Barcelona. F Rotondo and AC Ho-Palma were recipients of pre-doctoral fellowships of the Governments of Catalonia and Peru, respectively. The CIBER-OBN Research Web paid the salary of Dr. Romero.

### Competing Interests

The authors declare there are no competing interests.
## Author Contributions

- Floriana Rotondo performed the experiments, analyzed the data, authored or reviewed drafts of the paper, approved the final draft.
- Ana Cecilia Ho-Palma performed the experiments, analyzed the data, authored or reviewed drafts of the paper, approved the final draft.
- Xavier Remesar analyzed the data, prepared figures and/or tables, authored or reviewed drafts of the paper, approved the final draft, organized the work, compared the data and made calculations.
- José Antonio Fernández-López analyzed the data, prepared figures and/or tables, authored or reviewed drafts of the paper, approved the final draft, made most statistical analyses.
- María del Mar Romero performed the experiments, analyzed the data, prepared figures and/or tables, authored or reviewed drafts of the paper, approved the final draft, checked the methodology and quality of primers.
- Marià Alemany conceived and designed the experiments, prepared figures and/or tables, authored or reviewed drafts of the paper, approved the final draft.

## Animal Ethics

The following information was supplied relating to ethical approvals (i.e., approving body and any reference numbers):

The experimental design and the rat handling procedures were applied following the animal treatment guidelines established by the corresponding European, Spanish and Catalan Authorities. The Committee on Animal Experimentation of the University of Barcelona authorized the procedure used in this study (9443).

## Data Availability

Dipòsit Digital Institucional de la Universitat de Barcelona (University of Barcelona Repository)

http://hdl.handle.net/2445/119378.

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
