# Peer review of "Effect of sex on glucose handling by adipocytes isolated from rat subcutaneous, mesenteric and perigonadal adipose tissue"

_PeerJ, doi:10.7717/peerj.5440_

## Round 0.1 · original submission · Major Revisions

I agree with the general feeling from both reviewers that this work has merit. However, there are a number of issues with language and description of the work done that must be addressed before a final decision can be made. Please carefully address all the points of both reviewers.

·

Basic reporting

General,
The paper reports an experiment done to measure metabolic activity in adipose depots of rats. English is clearly not the first language of the authors, but it is basically acceptable. There are a few rough constructs that I have pointed out. Unfortunately the bigger issues are that the authors clearly do not know the literature, the truly massive and thorough literature on adipose tissue metabolism. A review published in 1980 (yes 1980, 38 years ago) in Progess in Lipid Research listed over 400 papers in adipose tissue metabolism, JUST IN RUMINANTS. There were hundreds more in rats. Finally, the completely incomprehensible statement of objectives precludes any ability to understand what they were doing and it was made even less understandable when the methods were given, which were absolutely inadequate to review as to what was actually done.
Abstract
Line 15: I don’t know what ‘canonical results’ means here, please define or reword
Line 57: “During decades’ is likely not the best translation in English. I might suggest “For most of the earlier research, the….”

Line 62; similar here, a segue is needed: “In the last twenty hears, many other functions of WAT have been discovered including synthesis of hormones and growth factors, insulin resistance in some diabetic states, interactions with other cell types that affect overall health. Included in these roles are the close control of….”
L 105, I am not sure of the precise anatomical definition of ‘inguinal cordons’…is there a different word? Epidydimal fat pads? Inguinal fat pads?
L 197 to 110 (the objective or hypothesis) I am very sorry to say but this is totally undefinable…it is impossible to tell what in fact the objective(s) or hypothesis(es) was or were.
‘which was the possible balance between the defining characteristics and relative metabolic uniformity”.
Balance between characteristics?
Balance between a characteristic and a metabolic activity?
What is a ‘relative metabolic uniformity’ and how can it be compared to anything else?
What is a ‘defining characteristic’ (anatomical location, fat cell size time of day….)

L 111. ‘influence of sex was included in the equation” what equation? Sex or gender?

This alone makes the work unable to be reviewed. It is impossible to know what you were testing. I will look over the rest quickly.

L 119 ‘acclimation’ to what? And did you measure feed intake or body weight?
L 153 to 157. Glucose, lactate, glycerol and NEFA were measured on what? With what sampling technique? On the cells? In the medium? Both?
L 175 to 178, as expected from the nebulous statement of objective, there is no clear statement of experimental design or statistical analysis. What ‘groups’ were compared and to what? Once the authors determine what their objectives and hypotheses were, and what treatments were applied and what was measured to test those objectives and hypothesis, they will need to clearly and explicitly state the statistical model used, including the proper error terms.
L 239-241, this is a massive assumption, as likely as it may be..without actually measuring incorporation. But release G3p from lipolysis could be reesterified.
L 244 there is absolutely no basis for this statement, without knowing the physiological and nutritional state of the animals when they were killed. If they were in energy and carbon balance it is obviousl that lipogenesis would be minimal, and in short term incubations there would not be sufficient time to induce all the enzymes and pathways to make significant amounts of fatty acids. You cannot use one set of animals in an undefined state to make such an inference.
L 255 same comment as above, the change in expression cannot absolutely predict the change in metabolic rate…it can only be supportive or consistent.
L 275 to 282, all of this is not surprising at all given what has been published many times on the amounts and transcriptional and posttranslational regulation of these enzymes which precludes a usual tight relationships (high CC or RSQ) between mRNA and activity.

L 305 to 309. This study does nothing of the sort, and unfortunately this paragraph simple demonstrates the authors ‘ignorance’ (and that is meant in the strict definition of the term..they clearly are not aware of the body of knowledge on WAT metabolism). Without sounding like a snide old man, it is likely that many of the authors were born after most of the work was done. Much of this would be found in works conducted from the 1960’s (or earlier) through the early 1980s. But to say ‘studies analyzing metabolic pathways are too few and incomplete to support it. Sir Hans Krebs, Veech, Per Belfrage, C. Holm, R. Vernon, D. Bauman, , RL Baldwin, R. Prior, Metz and Van den bergh, Abraham, Edwin Krebs, Katz, McNamara, Loor, etc…….might disagree with you.

I stopped reading here.
There is no way that they can possibly know glucose uptake or lactate uptake or output or NEFA or glycerol uptake given what they stated that they measured, as they did not state that they measured anything specifically in cells or media. AND in the absence of using some kind of radio or mass label, or measuring over time (0 minutes to XX minutes of incubation) how could they possibly know that ‘uptake’ and ‘release’ occurred compared to just recycling?

I think the authors collected much useful data and it would be nice to be published. But prior to redoing this paper with a new submission to this or any other journal, that they run a search at www.pubmed.com with the following search terms:
Title: adipose tissue metabolism
Date: 1950/01/01 to 1989/01/01.
Mine just now yielded 150 papers, several of which will be reviews and which cover all species including rats.

When they have a better command of the field and can make statements that demonstrate that knowledge, and when they can clearly describe their treatments, conditions an d

Experimental design

the review is above.

Validity of the findings

the review is above

Additional comments

the review is above

Reviewer 2 ·

Basic reporting

English must be improved. Also, the manuscript is lengthy and needs to be dramatically shortened.
Literature is fine.
Figures need to be re-arranged so that they are more understandable, shortened and obvious.
As present, the data might not support hypothesis.

Experimental design

The experimental design needs to be better described and more details are needed.

Validity of the findings

Any culture studies were conducted and thus, data need to be further confirmed.
Data need to be further verified.
Observed changes could also be due to cell damages and stresses experienced during adipocyte preparation, which needs to be better described.

Additional comments

Abstract: the background is not well described and sex difference was not even mentioned.
The language must be improved. In many cases, it is difficult to understand what were described.
Introduction:
It is lengthy and can be dramatically shortened.
Line 111: 3C needs to be defined.
Line 130: Method for mature adipocyte preparation is critical and more details need to be provided, in order for experiments to be replicable.
Line 131: Which type of collagenase was used? Collagenase with trypsin activity could cause significant damage to cell surface proteins, rendering them to behavior abnormally during culture.
Line 139: 30 ml/L , 3% serum, which is quite low. Why was it used?
Line 141: What are the relevant of glucose concentration? Why were 7 and 14 mM chosen?
Line 136: The microscopic views of prepared adipocytes need to be shown.
Line 148: Why chose 24 h or 48h? Wasn’t cells need certain time to recover from damage due to collagen treatment, centrifuge and other stresses during cell preparation? The difference between 24 h and 48 h cells could be due to cell adaptation to the in vitro culture system?
Line 445-449: These sentences are inaccurate. This study was done in cell culture, which does not contain mature adipose structure, and also the cell composition is different. Therefore, it is inaccurate to claim that adipocytes in vivo have similar changes observed in cultured cells.
Figure 2. It will be better to label significant difference in bars. Also, the variations for data of most figures were very small, which appeared unlikely for biological replicates. Do you use cell from individual animals and individual animal was considered as an experimental unit? These need to be added into the statistic section.
Figure 3 to 6. I found that these figures are lengthy, but provide little information. And frankly, difficult to understand. I would suggest to use tables or other ways to make them clear and concise. Also, I would like to see statistic difference to be labeled directly on data points.
Figure 7: Red circles need to be defined, such as representing cell membrane, mitochondria etc.
Metabolic pathways can be grouped into beta-oxidation, lipolysis, lipogenesis, and glycolysis, etc. And also needs to be properly labeled.
Overall, the manuscript is lengthy, and many sentences, words and explanations are unnecessary and then the trimmed off.
Language needs to be improved.

---

## Round 0.2 · Major Revisions

Several issues remain, including not properly addressing suggestions from the reviewers. Please ensure all comments are thoroughly addressed within the paper.

·

Basic reporting

First, thanks for the thorough response from the authors, it clarifies many things. I am sorry if I seemed too harsh , but whether we like it or not, the English language can be difficult and in in scientific writing there is even more difficulty of making things clear. We really do sometimes need to repeat things or to spell things out specifically, even though we think that the reader can connect one sentence to previous ones.
I also apologize if I criticized too harshly the understanding of the literature by the authors but as the paper was written, it was not at all clear that the authors had the knowledge of adipocyte metabolism, in any species (I just used ruminants as an example, there was a lot done in rats and pigs in the 'early days'). I think the new version is much improved.

I don't have any further comments other than the description of statistics is still not specific enough...you need to include the entire model statement(s).

I have no further comments and do not need to see any further revisions, I'll leave the final decisions to the Editor.

Experimental design

see above

Validity of the findings

see above

Additional comments

see above

Reviewer 2 ·

Basic reporting

In this revision, authors only addressed a portion of reviewers' comments. The manuscript remains long and lack of contents.
Figure 1 and 2 can be combined into one figure.
For other figures, the statistic tables below the figure panels are not informative and suggest to remove. Instead, it will be much better and clearer to label significant difference directly above bars. This comment has been proposed previously, but authors chose to ignore.

In addition, only in vitro data were conducted. Because adipocytes are severely stress (if they survive), the difference both 24 and 28 h data might just represent cell adaption to the new in vitro environment. As such, I think that presenting both 24 and 48 h data may not be meaningful.

Experimental design

Only in vitro data were present, which may not represent difference in vivo.

Validity of the findings

The validity of findings remain questionable.

Additional comments

Authors only address my comments partially.

---

## Round 0.3 · Minor Revisions

Thanks for addressing the previous comments. However, issues with language, in particular, still remain. Perhaps it is time for the authors to secure editorial help to address that issue.

Reviewer 2 ·

Basic reporting

This is the second revision, and I do not want to be too critical. In my opinion, the presentation of figures remains confusing.

Also, the English must be improved.

Experimental design

Only in vitro, which is a concern.

Validity of the findings

In my opinion, validity of the findings may be questionable, due to the lengthy adipocyte separation which likely altered cell behavior, among other problems.

I agree with authors that the manuscript will be difficult to further improve and, thus, I will leave it to the editor to make a decision.

Additional comments

The reviewer appreciate the effort of authors to improve the manuscript. However, there is only limited improvement.
The English must be improved.

---

## Round 0.4 · Minor Revisions

A final staff check has revealed that the English is still a bit idiosyncratic in places, therefore the journal is asking that you give it one final copyedit to improve the English grammar.

In addition, a few typos were noticed, for example:

L25: "convert" not "covert"
L26: "fatty acid"
L38: "triacylglycerol"

---

## Round 0.5 · accepted · Accept

No further changes needed.

#